Pathogens

# Coronavirus M protein disperses the trans-Golgi network and inhibits anterograde protein trafficking in the secretory pathway

Taylor M. Caddell[1], Rory P. Mulloy[2,3], Jennifer A. Corcoran[2,3], Eric S. Pringle[1]*, Craig McCormick[1]*

1 Department of Microbiology & Immunology, Dalhousie University, Halifax, Canada, 2 Department of Microbiology, Immunology and Infectious Diseases, University of Calgary, Calgary, Canada, 3 Snyder Institute for Chronic Diseases and Charbonneau Institute of Cancer Research, University of Calgary, Calgary, Canada

* eric.pringle@dal.ca (ESP); craig.mccormick@dal.ca (CM)

## Abstract

Coronaviruses (CoVs) encode a variety of transmembrane proteins that are translated and processed at the endoplasmic reticulum (ER). Three host ER resident transmembrane proteins, activating transcription factor 6 (ATF6), inositol-requiring enzyme 1 (IRE1), and PKR-like endoplasmic reticulum kinase (PERK), sense the accumulation of unfolded proteins in the ER and initiate the unfolded protein response (UPR) to maintain ER proteostasis. We observed that SARS-CoV-2 Spike broadly activated all three arms of the UPR, whereas the Membrane (M) protein selectively inhibited ATF6. ATF6 has a unique activation mechanism whereby ER stress triggers translocation to the Golgi where ATF6 is processed by resident proteases to release the ATF6-N transcription factor. We observed that M inhibited the stress-induced production of ATF6-N, suggesting that ATF6 failed to engage with Golgi proteases for processing. M also inhibited sterol regulatory element binding protein-2 (SREBP2)-mediated activation of sterol responses and stimulator of interferon response cGAMP interactor 1 (STING)-mediated activation of interferon responses, both of which are activated in the ER and require translocation to the Golgi for interactions that yield transcriptional responses. We observed that M accumulated in the cis-Golgi, and triggered dispersal of the trans-Golgi network (TGN). Using a cargo sorting assay, we determined that ER-to-Golgi cargo trafficking was intact in the presence of M, but cargo accumulated with M in the cis-Golgi and did not proceed further in the secretory pathway. We also observed aberrant cholesterol accumulation at the cis-Golgi with M, consistent with our observation of M association with detergent resistant membranes. Together, these data suggest that CoV M proteins interfere with Golgi architecture and trafficking. Because CoV egress does not require the canonical secretory pathway, this mechanism could allow the virus

**Data availability statement:** All relevant data are in the manuscript and its Supporting information files. Additional images for all biological replicates immunoblotting and immunofluorescence microscopy experiments are stored in the Dryad Database at the following link: DOI: 10.5061/dryad.rjdfn2zs6. Merged microscopy images were uploaded to https://amsterdamstudygroup.shinyapps.io/ezreverse/ for image inversion.

**Funding:** This work was supported by Canadian Institutes for Health Research (CIHR; https://cihr-irsc.gc.ca) Project Grant PJT-148727 (to C.M.), Coronavirus Variants Rapid Response Network (CoVaRR-Net; https://covarrnet.ca) Grant 175622 (to J.A.C. and others), and Nova Scotia COVID-19 Health Research Coalition Grants (https://researchns.ca/covid19-health-research-coalition/) to C.M. The funders had no role in study design, data collection and analysis, decision to publish, or preparation of the manuscript.

**Competing interests:** The authors have declared that no competing interests exist.

to selectively interfere with host responses to infection without impeding egress of nascent virions.

## Author summary

Coronaviruses (CoVs) use the endoplasmic reticulum (ER) for synthesis and processing of viral transmembrane proteins, including those that sculpt ER membranes into replication compartments as well as structural proteins that participate in virus assembly. We determined that amongst these CoV proteins, Spike triggered ER stress and activated all three arms of the unfolded protein response (UPR). By contrast, Membrane (M) selectively inhibited the ATF6 arm of the UPR, as well as two other ER-localized sensor proteins, SREBP2 and STING; all three sensors require anterograde trafficking in the secretory pathway to elicit downstream transcriptional responses. We demonstrated that M proteins accumulate with cholesterol at the *cis*-Golgi and inhibit anterograde trafficking from this site, while dispersing the *trans*-Golgi network (TGN). This property allows M to interfere with host responses to infection that depend on Golgi trafficking.

## Introduction

Coronaviruses (CoVs) are enveloped viruses with large, positive-sense, single stranded RNA genomes. Upon entry, the viral genome is released into the host cell cytoplasm and is immediately translated at the endoplasmic reticulum (ER) into long polyproteins that are processed by two viral proteases, generating a collection of non-structural proteins (Nsps) [1]. Among these, Nsp3, Nsp4, and Nsp6 are ER-localized transmembrane proteins required for formation of ER-derived double-membrane replication organelles (ROs) [2–4], the site of viral RNA (vRNA) synthesis [5]. CoV transmembrane structural proteins, Spike, Membrane (M), and Envelope (E), are translated at the ER and traverse the secretory pathway for further modifications. Spike is a large type I transmembrane protein that is activated by site-specific proteolysis to generate the S1 attachment subunit and the S2 fusion subunit [6,7]; this processing can be mediated by Golgi-resident proteases, or uncleaved Spike can be incorporated into viral particles and post-egress cleavage can be mediated by cell surface or lysosomal proteases during viral entry [6,7]. E is a small type I transmembrane protein that forms pentamers and functions as a viroporin, increasing the lumenal pH in the Golgi and protecting Spike from excessive proteolysis in this compartment [8]. M, the most abundant protein in the viral envelope, is a multi-spanning membrane protein, with a small lumenal amino-terminus, three transmembrane domains, and a large cytoplasmic carboxy-terminal domain known as the endodomain [9]. Ectopically expressed M displays steady-state accumulation in the Golgi [10,11], but it can also be retrieved to earlier compartments, including the ER. This bidirectional traffic is key to its role in virus assembly and budding at the

ER-Golgi Intermediate Compartment (ERGIC), allowing M to capture other structural proteins and redirect them to assembly sites, even after they have trafficked to the *trans*-Golgi. These interactions are largely governed by the endodomain, which interacts with the other structural proteins (Spike, E, and nucleocapsid (N)) [12–14]. Furthermore, M-M interactions are thought to drive membrane curvature and exclusion of host proteins during envelope formation [15]. E assists M in controlling the trafficking and processing of Spike, and as assembly takes place, E acts as an enhancer of budding. Together, these structural proteins, along with N, coordinate assembly of CoV virions at the ERGIC followed by lysosomal egress and release [16].

ER proteostasis could be perturbed by bursts of viral protein synthesis at the ER and remodeling of ER membranes into ROs. Three ER-localized transmembrane proteins are activated in response to the accumulation of unfolded proteins in the ER lumen: activating transcription factor 6 (ATF6), inositol-requiring enzyme 1 (IRE1), and PKR-like endoplasmic reticulum kinase (PERK). Activation of these proteins leads to the generation of basic leucine zipper (bZIP) transcription factors (TFs) that orchestrate the unfolded protein response (UPR) by transactivating genes involved in increasing ER protein folding and processing capacity [17,18]. ER stress triggers ATF6 translocation from the ER to the Golgi where it is cleaved by resident proteases, liberating the cytosolic amino-terminal ATF6 fragment comprising the ATF6-N bZIP TF [19–21]. By contrast, in response to ER stress, IRE1 generates a bZIP TF via a cytosolic endonuclease domain that cleaves *X-box binding protein 1* (*XBP1*) mRNA at two sites, removing a small intron and shifting an open reading frame to generate the XBP1s bZIP TF [22–25]. Finally, ER stress triggers PERK-mediated eIF2α phosphorylation [26], limiting guanine nucleotide exchange on eIF2 [27,28], and favouring upstream open reading frame (uORF) skipping on the *ATF4* mRNA required for synthesis of the ATF4 bZIP TF [29]. Accumulating evidence indicates complex interactions between CoVs and the UPR, whereby infection activates UPR sensors [30–35], but with attenuated transcription of UPR-responsive genes [36]. While our understanding of viral factors that regulate the UPR remains incomplete, several viral UPR agonists have been identified including Spike proteins from diverse CoVs [32,37,38], as well as SARS-CoV transmembrane proteins Nsp6 [39], ORF3a [40], and ORF8ab [41] proteins. The presence of UPR agonists and muted downstream responses suggests that CoVs encode UPR antagonists.

In this study, we aimed to identify SARS-CoV-2 proteins that modulate the UPR, specifically screening for antagonists of UPR signaling. We determined that M selectively inhibits the ATF6 branch of the UPR by inhibiting anterograde protein trafficking in the secretory pathway. We demonstrate that M accumulates with cholesterol at the *cis*-Golgi and inhibits anterograde trafficking from this site, while dispersing the *trans*-Golgi network (TGN). This mechanism could allow the virus to selectively interfere with host responses to infection without impeding egress of nascent virions.

## Results

**Coronavirus proteins modulate ATF6-dependent transcription.** To screen for antagonists of UPR activation we used a luciferase reporter containing an ER stress response element (ERSE) promoter to measure ATF6 activation. ATF6 responds to proteotoxic stress resulting in translocation from the ER to the Golgi where it is cleaved to liberate the cytosolic amino-terminal bZIP TF called ATF6-N, which migrates to the nucleus and binds to ERSEs to transactivate genes encoding chaperones, foldases, and lipogenesis factors. We co-transfected our ERSE-luciferase reporter, plasmids from a SARS-CoV-2 open reading frame (ORF) library [42], and a *Renilla*-expressing plasmid for normalisation, to measure ATF6-N-dependent transcriptional responses. We observed that four SARS-CoV-2 proteins significantly increased ATF6-N-dependent firefly luciferase activity: Spike, Nsp4, ORF8 and ORF10 (Fig 1A). These findings are consistent with previous reports of ATF6 activation by Nsp4 [43] and Spike activation of the UPR, but not ATF6 specifically [32,37,38]. Because the ORF library we used featured C-terminal 2X-Strep tags that might interfere with protein function, we also tested an untagged Spike; this construct similarly, yet more potently, activated ATF6 (Fig 1B). We used this untagged Spike construct as an agonist in our counter-screen for SARS-CoV-2 proteins that inhibit ATF6-N-dependent

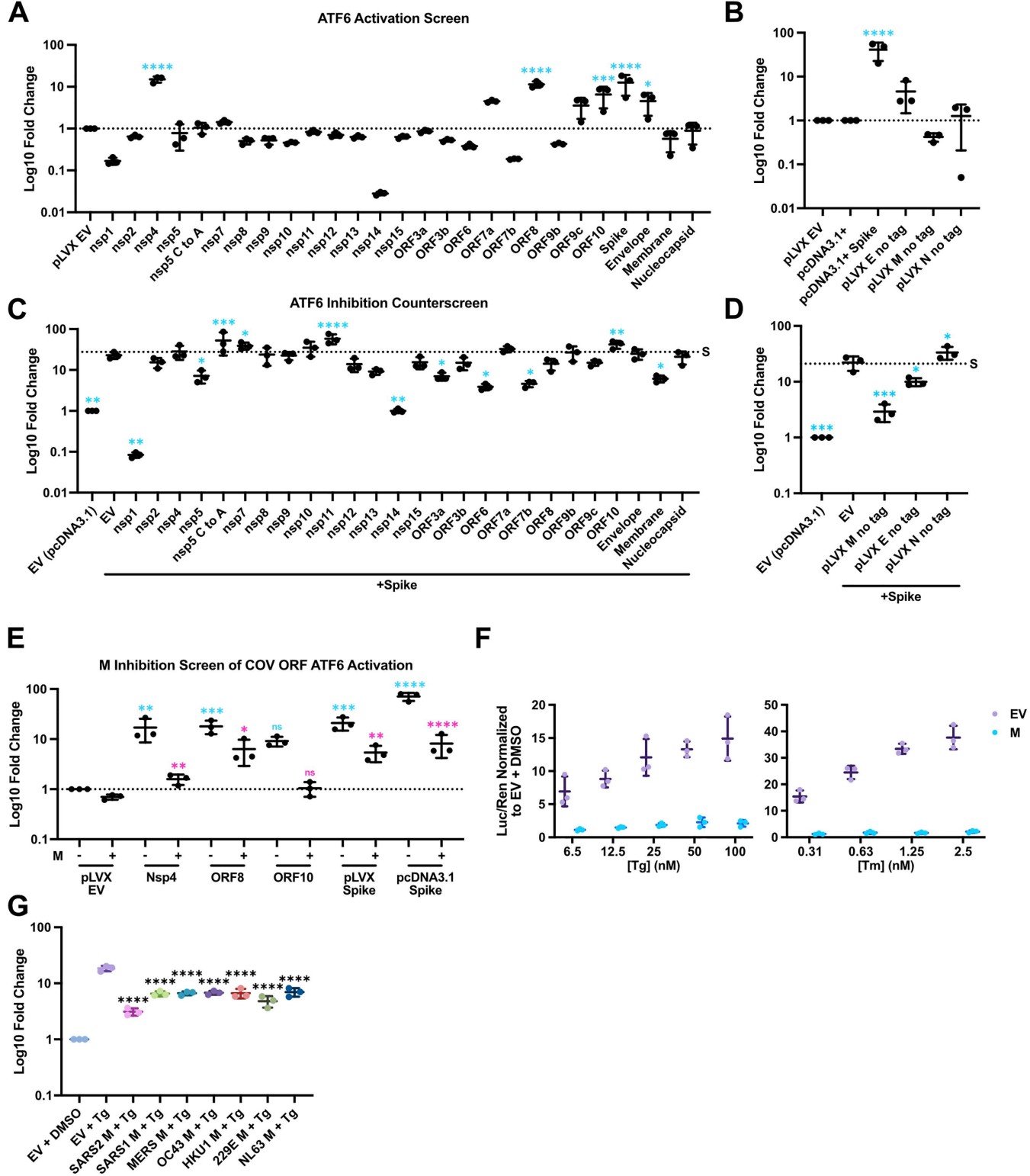

**Fig 1. Coronavirus proteins modulate ATF6-dependent transcription. (A-E)** SARS-CoV-2 ORF screen for ATF6 activators and inhibitors. HEK293T cells were co-transfected with pLVX vectors encoding viral ORFs plasmids or pLVX Empty Vector (EV) with ERSE-*luc* and CMV-*Renilla* reporters for 24h

prior to harvest for luciferase assay. **(A)** SARS-CoV-2 ORF library where all ORFs encode a C-terminal 2x-Strep tag. **(B)** As in A with untagged SARS-CoV-2 structural proteins. **(C)** A counter-screen for inhibitors was conducted as in **(A)** with the addition of Spike as an ATF6 agonist. **(D)** As in **(B)** with the addition of Spike as an ATF6 agonist. **(E)** as in **(D)** with the addition of M as an ATF6 antagonist as indicated. Blue asterisk indicate significance relative to EV, pink asterisks indicate significance of M co-transfection relative to paired viral ORF alone **(F)** HEK293T cells were transfected with M then treated as indicated with thapsigargin (Tg) or tunicamycin (Tm) the following day for 24h prior to harvest at 48h. **(G)** as in **(F)** where cells were transfected with M proteins from seven coronavirus known to infect humans or EV. (n = 3 ± SD, statistical significance was determined by one-way ANOVA with Fisher's LSD test. *, adjusted P value < 0.05; ***, adj. P < 0.0002; and ****, adj. P < 0.00001 relative to EV.).

ERSE-luciferase activity; we identified seven such proteins: Nsp1, Nsp5, Nsp14, ORF3a, ORF6, ORF7b, and M (Fig 1C). Among these, Nsp1 and Nsp14 are known host shutoff proteins and should therefore inhibit expression of luciferase from the reporter constructs by interfering with host mRNA processing and translation [44–49]. We selected M for further investigation because of its well-documented regulation of Spike trafficking and virus assembly at the ERGIC [14] and its conservation across the *Coronaviridae*. An untagged version of the M construct similarly inhibited Spike-mediated ATF6 activation (Fig 1D).

To determine whether M possesses broader ATF6 inhibiting activity, we tested M in combination with other ATF6-activating SARS-CoV-2 proteins and observed that M suppressed ATF6 activation by Nsp4 and ORF8 (Fig 1E). M also inhibited ATF6 activation by chemical agonists Thapsigargin (Tg), which triggers ER stress by inhibiting the sarcoplasmic/endoplasmic reticulum calcium ATPase 2 (SERCA2) [50], and Tunicamycin (Tm), which causes ER stress by inhibiting N-linked glycosylation leading to improper protein folding [51] (Fig 1F). We tested M proteins from seven human CoVs for the ability to inhibit ATF6 activation by Tg and found this ability was conserved (Fig 1G). We confirmed the specificity of our ATF6 reporter assay by demonstrating that neither Spike nor Tg could stimulate ERSE-luciferase activity in ATF6 knock-out (KO) cells (S1 Fig). Together, these findings demonstrate that M broadly inhibits ATF6 activation, and that this function is conserved in M proteins from multiple human CoVs.

**SARS-CoV-2 M protein inhibits ATF6 activation but not PERK or IRE1 activation.** ATF6 activation requires anterograde transport and cleavage by Golgi-resident proteases to generate the ATF6-N bZIP transcription factor, unlike IRE1 and PERK that elicit the synthesis of bZIP transcription factors following phosphorylation events at ER membranes (Fig 2A). We next evaluated whether Spike and M could regulate other branches of the UPR by overexpressing Spike, E, and M alone and used immunoblotting to detect activation of UPR sensors. Treatment of cells with Tg activated all three UPR branches, with (i) PERK activation indicated by the presence of a slow-migrating phosphorylated PERK (indicated with *) and increased production of the ATF4 target CHOP, (ii) IRE1 activation indicated by accumulation of XBP1s, and (iii) ATF6 activation indicated by increased levels of BiP (Fig 2B). Ectopic expression of Spike caused activation of all three branches of the UPR as well, with E also promoting PERK phosphorylation. Neither E nor M had any appreciable impact on the IRE1 and ATF6 branches. We confirmed that M inhibited Spike-mediated ATF6 activation via diminished accumulation of BiP protein compared to control (Fig 2C). Conversely, M had no effect on Spike-mediated activation of PERK (PERK phosphorylation and CHOP accumulation), or IRE1 (XBP1s accumulation). Furthermore, M did not inhibit accumulation of spliced XBP1 mRNA (Fig 2D). This indicates that while Spike activates all three branches of the UPR, M appears to only limit activation of ATF6. ATF6 activation requires translocation from the ER to the Golgi to access proteases that cleave and release the ATF6-N TF. We assessed cleavage of ATF6 using an HA-epitope tagged construct (HA-ATF6) via immunoblotting and found that M suppressed accumulation of the N-terminal cleavage product during Tg treatment and basal turnover in untreated cells (Fig 2E). Consistent with this, in ATF6 knockout (KO) cells, an ectopically expressed ATF6-N construct that bypasses the secretory pathway potently activated the ERSE-luciferase reporter, even in the presence of M (Fig 2F). Taken together, this suggests that M selectively inhibits the ATF6 branch of the UPR by impeding ATF6 translocation to the Golgi or cleavage by resident proteases during stress.

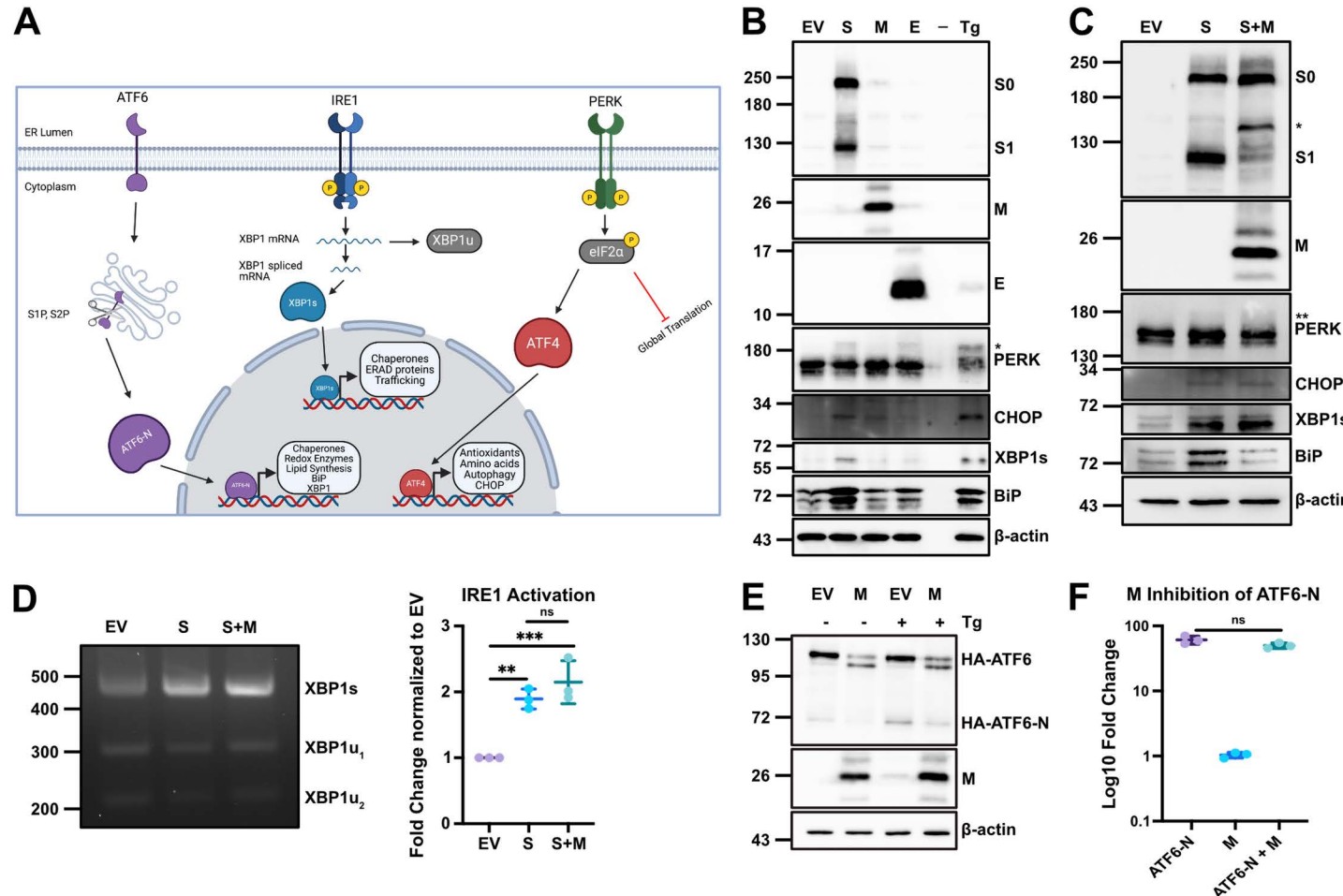

**Fig 2. SARS-CoV-2 M protein inhibits ATF6 activation but not PERK or IRE1 activation. (A)** Diagram of the unfolded protein response (UPR) in brief displaying the intermediate steps that occur for activation and generation of transcription factors that lead to individual transcriptional responses. Created in BioRender. Caddell, T. M. (2026) https://BioRender.com/yuwuyag. **(B-C)** HEK293T cells were transfected with SARS-CoV-2 structural proteins as indicated for 24h before lysate was harvested for western blotting. As a positive control untransfected cells were treated with 200 nM thapsigargin (Tg) for 4h prior to harvest. **(D)** as in **(B)** except RNA was harvested for RT-PCR for *Xbp1* mRNA splicing assay. **(Left)** agarose gel; **(Right)** quantification of image by densitometry. **(E)** as in **(B)** where cells were transfected with HA-ATF6 with either M or EV control. Cells were treated with 200 nM Tg treatment for 4h prior to harvest as indicated. **(F)** HEK293T cells were co-transfected with ERSE-*luc*, CMV-*Renilla*, and HA-ATF6-N with or without M as indicated for 24h prior to harvest for luciferase assay. (For blots, representative image of n=3 is shown. For graphs, n=3±SD, statistical significance was determined by one-way ANOVA with Fisher's LSD test. ns – not significant; *, adjusted P<0.05).

**SARS-CoV-2 M protein inhibits activation of host pathways that require anterograde transport in the secretory pathway.** We investigated if other host transmembrane proteins that are similarly activated at the ER and subsequently transported to the Golgi were inhibited by M. Sterol regulatory element binding protein-2 (SREBP2) activation shares a similar ER to Golgi activation pathway as ATF6 and releases an amino terminal cytosolic portion of the protein comprising the bZIP TF SREBP2-N [52]. Using a reporter construct bearing three sterol response elements (SREs) that direct transcription of a firefly luciferase gene [53], we tested the effects of M on sterol responses. Sterol depletion of HEK293T cells by (i) serum starvation or (ii) treatment with cerivastatin (ST) that inhibit *de novo* sterol synthesis via 3-Hydroxy-3-methylglutaryl-CoA reductase inhibition, both increased SRE-luciferase activity (Fig 3A). By contrast, cells that

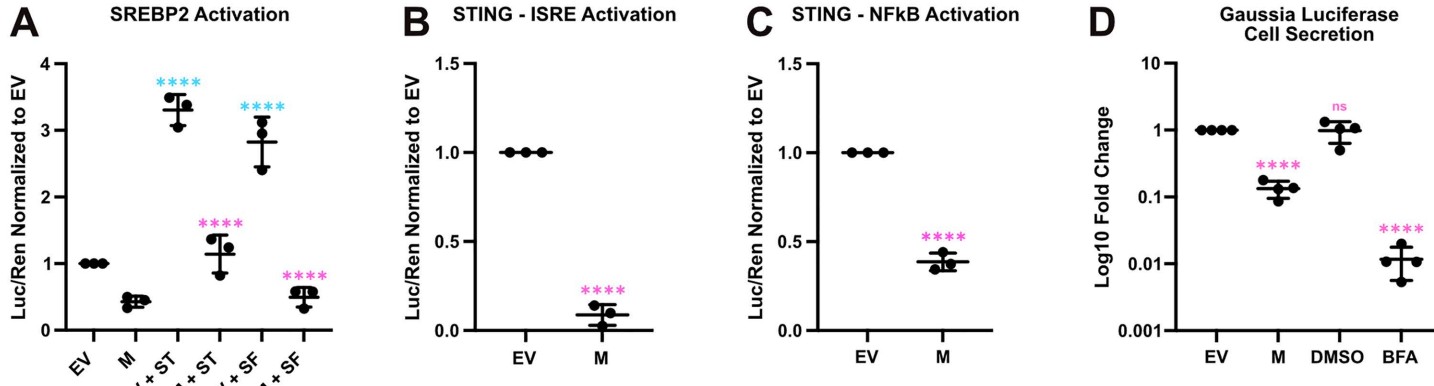

**Fig 3. SARS-CoV-2 M protein inhibits activation of host pathways that require anterograde transport in the secretory pathway. (A)** HEK293T cells were co-transfected with LDLR-*luc*, CMV-*Renilla*, and M or EV as indicated. The following day cells were treated with 10 μM cerivastatin (ST) or placed in serum-free media (SF) for 24 h prior to harvest for luciferase assay. **(B-C)** HEK293T cells were co-transfected with a **(B)** ISRE-*luc* or **(C)** NFkB-*luc, along* with cGAS and STING encoding plasmids (to stimulate ISRE and NFkB activation) and with either M or EV as indicated. Cells were harvested at 24 h for luciferase assays. **(D)** HEK293T cells were transfected with secreted *Gaussia* luciferase and EV or M as indicated. 24 h after transfection, medium was removed and replaced with fresh media. As a positive control cells were treated with 10 μM Brefeldin A (BFA) or DMSO vehicle control for 6h as indicated. Both supernatant and cells were then harvested at 30 h post-transfection for luciferase assay. (n = 3 ± SD or n = 4 ± SD **(D)**), statistical significance was determined by one-way ANOVA with Fisher's LSD test. ns – not significant; *, adjusted P < 0.05; ****, adj. P < 0.00001 relative to EV).

expressed SARS-CoV-2 M protein were limited to baseline SRE-luciferase activity in response to sterol depletion. We also investigated the function of the STING pathway. At rest, STING is an ER-resident homodimeric transmembrane protein [54]. Binding to cyclic GMP-AMP (cGAMP) triggers a conformational switch in STING homodimers causing anterograde transport to the Golgi [54,55], it is then phosphorylated by TANK-binding kinase 1 (TBK1) in the *trans*-Golgi network (TGN) [56] and drives interferon regulatory factor 3- and NF-κB-dependent transcription [57,58]. To test STING pathway function, we ectopically co-transfected HEK293T cells with cGAS and STING constructs, as well as firefly luciferase reporter constructs bearing interferon-stimulated response elements (ISREs) or NF-κB response elements. cGAS/STING co-transfection activated ISRE-luciferase (Fig 3B) and NF-κB-luciferase (Fig 3C) reporters as expected. By contrast, co-transfection of M with these complexes caused a marked decrease in output from both reporters. Thus, in addition to ATF6, M also inhibits the SREBP2 and cGAS/STING pathways, all of which traverse the Golgi to execute their respective downstream transcription programs. Finally, to determine whether M more generally affects the trafficking of proteins in the secretory pathway, we transfected cells with a secreted *Gaussia* luciferase construct followed by harvest of cell lysates and cell supernatants for luciferase assays. Brefeldin A, a drug that inhibits the GBF1 guanine nucleotide exchange factor for Arf1-GTPase, thereby inhibiting vesicular transport between the ER to the Golgi and causing fusion of ER and Golgi membranes [59–61], dramatically reduced secretion of the *Gaussia* enzyme (Fig 3D). M also greatly diminished *Gaussia* secretion. Thus, M causes broad defects in anterograde transport in the secretory pathway for transmembrane and soluble lumenal protein cargo.

**Coronavirus M proteins localize to the *cis*-Golgi and disperse the structure of the *trans*-Golgi network.** With evidence for M-mediated inhibition of anterograde protein trafficking, we investigated the position of M in the secretory pathway. Consistent with previous observations, we observed that ectopically expressed SARS-CoV-2 M accumulated in the *cis*-Golgi [10,11], as indicated by co-localization with Golgi matrix protein 130 kD (GM130), a peripheral membrane protein of the *cis*-Golgi involved in maintaining organelle structure [62] (Fig 4A). By contrast, we did not observe steady-state co-localization between M and ERGIC-localized lectin ERGIC53 in HEK293T cells stably expressing green fluorescent protein (GFP)-ERGIC53 fusion protein (GFP-ERGIC53) (Fig 4A and 4B), or with the *trans*-Golgi Network (TGN) marker

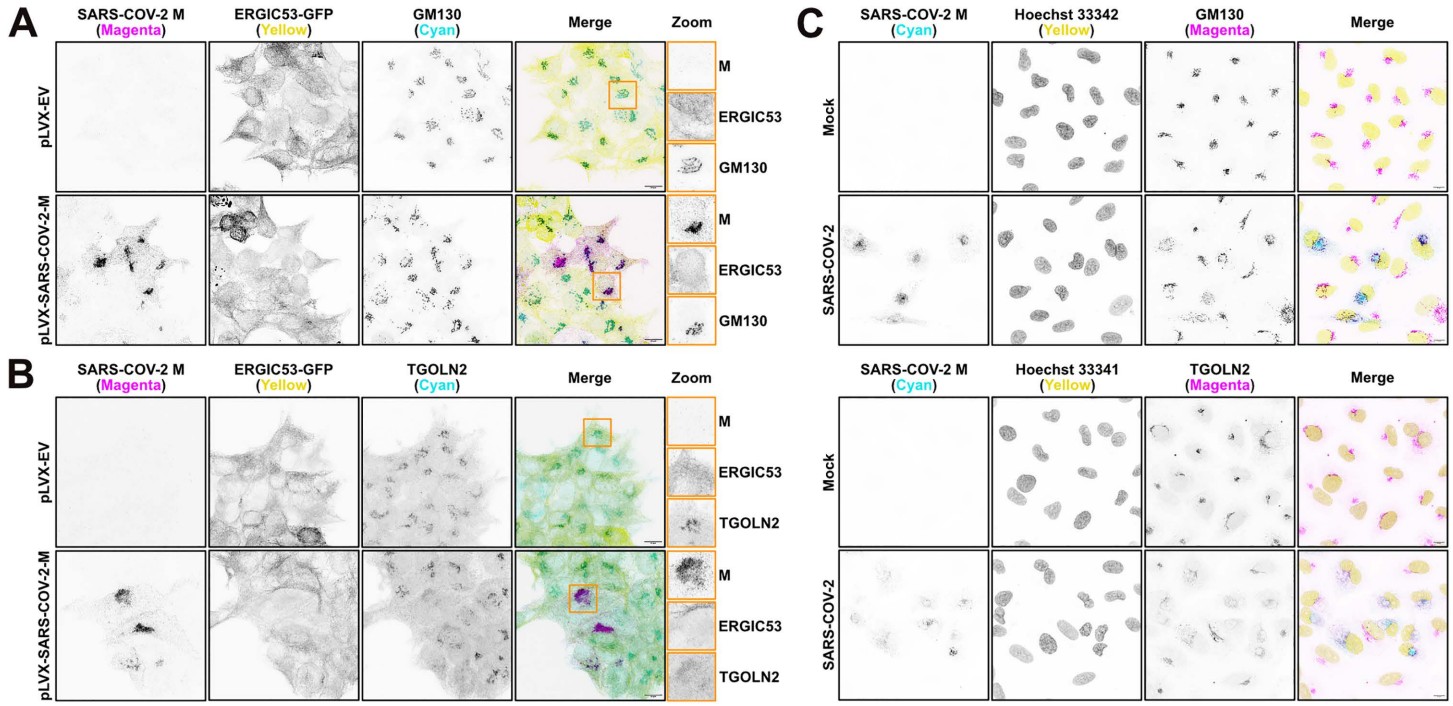

**Fig 4. SARS-CoV-2 M protein localizes to the *cis*-Golgi and disperses the structure of the *trans*-Golgi network.** ERGIC53-GFP-HEK293T cells were transfected with M or EV control and fixed 24h later **(A-B)**. ACE2-A549 cells were infected with SARS-CoV-2 at an MOI of 1 then fixed at 24 h post-infection **(C)**. Cells were immunostained as indicated with antibodies targeting SARS-CoV-2 M, GM130 (*cis*-Golgi), or TGOLN2 (*trans*-Golgi) and imaged by confocal microscopy. Maximum intensity projections are presented. 100X magnification, scale bar = 10 μm, Orange boxes indicate zoomed field of view. Representative images of three independent experiments.

TGOLN2/TGN46 (Fig 4B), supporting predominant accumulation of M in the *cis*-Golgi. We also observed TGN dispersal and reduced staining for TGOLN2 in cells expressing M, suggesting that this compartment may be disrupted.

SARS-CoV-2 and other HCoV infections have been shown to cause Golgi fragmentation and dispersal [63–66]. These studies used TGOLN2/TGN46 as a Golgi marker for dispersal, which we have also found to disperse in the presence of SARS-CoV-2 M. As we identified an accumulation of SARS-CoV-2 M at the *cis*-Golgi indicated by GM130, we wanted to further investigate how the *cis*-Golgi and TGN appear during infection. We infected A549-ACE2 cells with SARS-CoV-2 and immunostained with an anti-M antibody along with markers for *cis*-Golgi (GM130) and TGN (TGOLN2), which revealed that SARS-CoV-2 M accumulates in the *cis*-Golgi during infection, with a fragmented staining pattern for both the *cis*-Golgi and the TGN compared to uninfected cells (Fig 4C). We also observed fragmentation and dispersal for both the *cis*-Golgi and the TGN in HCoV-OC43 and HCoV-229E infections (S2 Fig), indicating that this phenotype is conserved amongst divergent HCoVs.

**SARS-CoV-2 M protein restricts protein trafficking in the Golgi.** We used the Retention Using Selective Hooks (RUSH) cargo sorting assay [67] to measure the effects of SARS-CoV-2 M on dynamic protein trafficking in the secretory pathway in living cells (Fig 5A). This assay employs a two-component system comprised of an ER-restricted 'hook' construct that includes a streptavidin domain, and a 'reporter' construct that includes a streptavidin binding peptide (SBP) fused to a fluorescent protein. As these proteins accumulate in the cell they co-localize in the ER via streptavidin-SBP interactions. Biotin addition competes with this interaction and releases the SBP-containing reporter construct from the 'hook', allowing synchronous trafficking of reporter proteins through the secretory pathway. We used a type II ER-retained

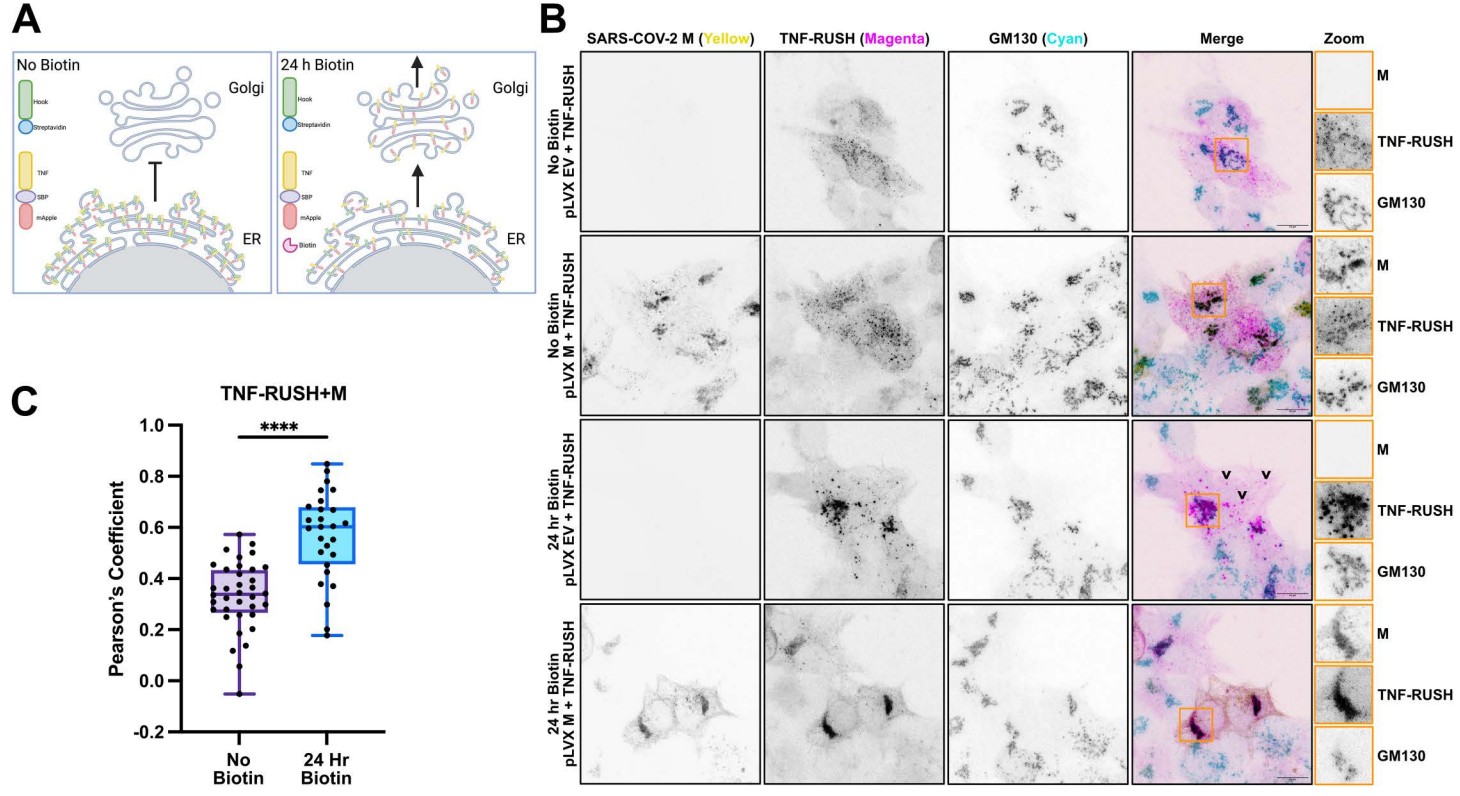

**Fig 5. SARS-CoV-2 M protein restricts protein trafficking in the Golgi. (A)** Model of Retention Using Selective Hooks (RUSH) system. TNF-mApple with a strepavidin-binding peptide (SBP) is held in the ER by ER-retained streptavidin. Biotin treatment out-competes SBP-strepavidin interaction and allows TNF-mApple to continue anterograde traffic in the secretory pathway. Created in BioRender. Caddell, T. M. (2026) https://BioRender.com/3fvxi3v. **(B)** HEK293T cells were co-transfected with mApple TNF-a-RUSH with either M or EV control. Cells were then treated with 50 μM biotin for 24h, as indicated. Cells were then fixed 24h post-transfection and stained with antibodies targeting SARS-CoV-2 M, and GM130 (*cis*-Golgi), and imaged by confocal microscopy. **(C)** Pearson's coefficient measuring signal overlap between M and TNF-RUSH channels with or without biotin treatment. Maximum intensity projections are presented. 100X magnification, scale bar = 10 μm, Orange boxes indicate zoomed field of view. Representative images of three independent experiments. A two-tailed unpaired Welch's t-test was performed, **** = adjusted p value <0.0001.

Streptavidin-Ii hook protein and a type II transmembrane tumor necrosis factor (TNF) protein domain fused to SBP and the mApple fluorescent protein [68]. We observed that the reporter construct accumulated in the ER as expected, both in the presence of M or empty vector (EV) control, with punctate structures representing previously described ER exit sites (ERESs) [68] and no signal overlap with the *cis*-Golgi compartment (Fig 5B). After 24 h of biotin treatment in EV transfected control cells, the reporter protein formed large puncta that partially co-localized with the *cis*-Golgi marker, displaying proper translocation and trafficking through the secretory pathway, as well as more distal puncta closer to the cell periphery (indicated by black arrows, Fig 5B), indicative of vesicular trafficking toward the cell surface [67]. In cells expressing SARS-CoV-2 M, after biotin treatment, the bulk of the reporter protein was found in the *cis*-Golgi, strongly overlapping with M localization, and little to no distal puncta observed. Quantitative analysis confirmed significant co-localization of TNF-RUSH and M following biotin-mediated release of the TNF-RUSH cargo from the ER compartment (Fig 5C). These data suggest that while M does not interfere with anterograde trafficking of host proteins from the ER to the *cis*-Golgi, further progress through the secretory pathway beyond the *cis*-Golgi is stymied.

**SARS-CoV-2 M protein is found in detergent resistant membranes and recruits cholesterol.** Efficient protein trafficking in the Golgi requires proper distribution of cholesterol and sphingolipids in Golgi membranes. The COPI coatomer

PLOS Pathogens

complex that mediates intra-Golgi transport and Golgi-to-ER retrograde transport is recruited to Golgi membranes by Arf GTP-binding proteins and orchestrates vesicle formation, cargo sorting, vesicle scission, and vesicle uncoating [69]. In the presence of Arf1-GTP, COPI coatomer complexes partition into liquid-disordered domains and are excluded from liquid-ordered domains rich in sphingolipids and cholesterol, resulting in the formation of vesicles bearing far fewer of these lipids than the Golgi compartments from which they derive [70]. Accordingly, these COPI-mediated transport mechanisms are sensitive to perturbations in levels of sphingolipids and cholesterol in the Golgi [71]. We hypothesized that M inhibits anterograde secretion by altering cholesterol abundance in the *cis*-Golgi. To determine whether M is present in liquid-ordered domains, we isolated detergent resistant membranes (DRMs) from cells transfected with EV or M and conducted immunoblotting for cellular targets. M was found to be present in both the top (DRMs) and bottom (detergent soluble membranes (DSMs)) fractions (Fig 6A). Calnexin was selected as a marker for DSMs and Caveolin was selected as a marker for DRMs. Spike, which is heavily palmitoylated, was previously shown to increase the ordered lipid environment

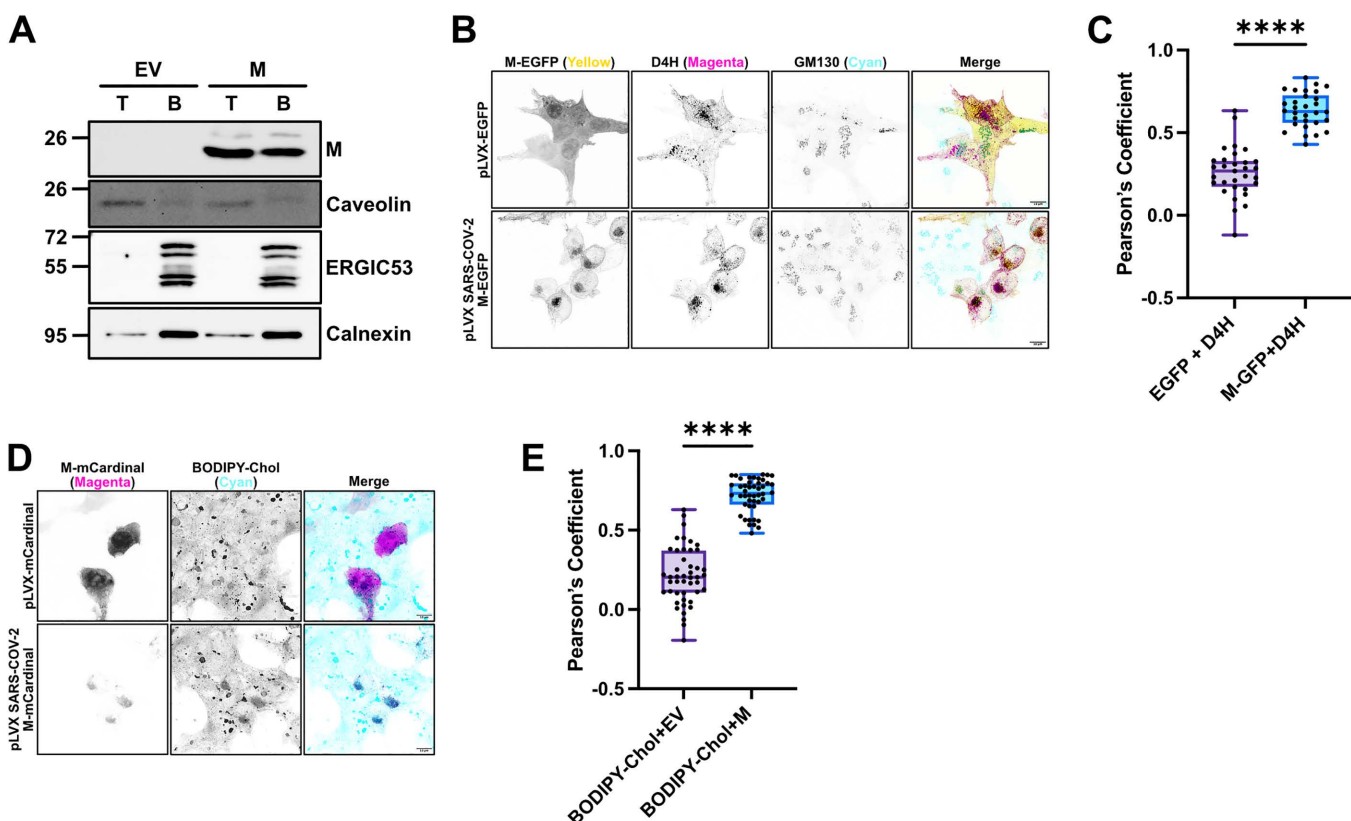

**Fig 6. SARS-CoV-2 M protein is present in detergent resistant membranes and recruits cholesterol. (A)** HEK293T cells were transfected with M or EV and DRMs were isolated as described in methods. The top fraction, containing the DRMs and the bottom fractions were used for immuno blot analysis as indicated. A representative image of n = 3 is shown. **(B)** HEK293T cells were co-transfected with cholesterol biosensor D4H-mCherry and pLVX-EGFP or pLVX-M-EGFP. Cells were fixed at 24h and imaged without detergent permeabilization. **(C)** Pearson's Correlation Coefficient measuring signal overlap between EGFP and D4H channels in pLVX-EGFP or pLVX-M-EGFP conditions from panel **(B)**. **(D)** HEK293T cells were transfected with pLVX-mCardinal or pLVX-M-mCardinal. At 4h post-transfection, medium was replaced with 10% FBS DMEM containing BODIPY 480/508-cholesterol. Cells were fixed at 24h and imaged without detergent permeabilization. **(E)** Pearson's Correlation Coefficient measuring signal overlap between BODIPY-cholesterol and pLVX-mCardinal or pLVX-M-mCardinal channels from panel **(D)**. Images were taken at 100X magnification, scale bar = 10 μm. For panels **(C)** and **(E)** each data point is one cell evaluated for signal overlap across multiple fields of view and 3 independent biological replicates. A two-tailed unpaired Welch's t-test was performed, **** = adjusted p value <0.0001. Representative images of three independent experiments.

of the ERGIC causing accumulation of the ERGIC53 lectin in DRMs [72]. By contrast, M had no obvious effect on the lipid composition of the ERGIC, with ERGIC53 remaining associated with DSMs in M-expressing cells.

DRMs are membrane microdomains enriched in cholesterol and sphingolipids. To determine whether the presence of M in DRMs reflected alterations in cholesterol levels in the *cis*-Golgi, we co-transfected HEK293T cells with M-EGFP or EV along with a D4H-mCherry cholesterol biosensor [73]. We observed that cholesterol distribution in EV control cells showed no discernable overlap with the *cis*-Golgi marker GM130, as expected (Fig 6B). By contrast, cells expressing M-EGFP displayed strong overlap between M-EGFP, D4H-mCherry, and GM130; M co-localization with the D4H probe was confirmed by a Pearson's correlation coefficient test (Fig 6C). We confirmed this observation using the BODIPY 480/508-cholesterol (BCh2) probe [74]; we transfected cells with mCardinal or M-mCardinal and observed significant co-localization of BCh2 with M-mCardinal, but not the mCardinal control (Fig 6D and 6E). Therefore, we concluded that SARS-CoV-2 M localizes to the *cis*-Golgi, which correlates with recruitment of cholesterol to this site and the association of M with DRMs.

## Discussion

CoVs encode proteins that limit host antiviral responses by preventing the detection of viral products by host sensors, and if viral infection is detected, additional viral proteins prevent the synthesis of host antiviral proteins. Here we describe another reinforcing layer of viral manipulation focused on the host secretory pathway. During infection, CoVs use the ER for synthesis and processing of viral transmembrane proteins, including those that sculpt ER membranes into replication compartments as well as structural proteins that participate in virus assembly. We reasoned that these dramatic ER perturbations could affect ER proteostasis and the UPR. We observed UPR modulation by numerous CoV proteins, including Spike, which broadly activated all three arms of the UPR. By contrast, M selectively inhibited the ATF6 arm of the UPR by preventing cleavage and release of the ATF6-N TF from the Golgi. We found that M more generally inhibited ER-to-Golgi transport, including SREBP2-mediated activation of sterol responses, STING-mediated activation of downstream signaling responses and bulk secretion of *Gaussia* luciferase. Using a RUSH cargo sorting assay, we observed that M accumulated in the *cis*-Golgi and inhibited further anterograde transport of a transmembrane reporter protein beyond this compartment, while also dispersing the TGN, a phenotype that we also observed during infection with diverse CoVs. Together, these observations suggest that CoV M protein disrupts the TGN and impedes normal anterograde traffic in the canonical secretory pathway. Because CoV egress does not require the TGN, this mechanism could allow the virus to selectively interfere with host responses to infection without impeding egress of nascent virions. This research adds yet another mechanism of host subversion to the impressive CoV armamentarium.

Golgi dispersal during viral infection is commonplace, but precise mechanisms and biological significance are not always clear [75]. Our observations of Golgi dispersal during CoV infection is consistent with several previous reports [63–66]. One group identified multiple SARS-CoV-2 proteins that are sufficient to trigger Golgi dispersal, including M [64]. By contrast, using immunostaining and a cargo sorting assay, we observed accumulation of M at the *cis*-Golgi, which correlated with a failure of diverse cargo proteins to traffic beyond this compartment. Our observations are consistent with accumulating evidence that Golgi dispersal during viral infection impedes protein trafficking in the secretory pathway.

How does M impede protein cargo trafficking in the secretory pathway, and could it relate to its role in virion assembly? M is a polytopic transmembrane glycoprotein, with a short lumenal N-glycosylated amino-terminal domain, three transmembrane helices connected by short loops, and a cytoplasmic hinge region that connects to an extended carboxy-terminal β-sheet sandwich domain (BD) [9]. M is the most abundant viral structural protein, and virion assembly places the short amino terminal domain on the exterior of the virion, the transmembrane domains in the virion envelope, and the extended carboxy-terminal domain in the virion interior, where it interacts with the viral genome. Structural studies have revealed the importance of M dimer formation and the formation of higher-order oligomers in the virus assembly process [9,76,77]. Cryo-electron microscopy studies demonstrate that M forms homodimers where three-helix transmembrane domain bundles are comprised of helices from different monomers, and lateral interactions of these dimers

in the membrane can lead to tetrameric and hexameric oligomers. Furthermore, these M dimers exist in two forms dictated by conformational changes in the hinge region, a "short" form with a compact appearance, and a "long" form with an extended appearance. Both forms of M are required for virus assembly and can be found in intact virions [9], with the short form associated with membrane flexibility and low Spike density, and the long form associated with more rigid membranes and Spike clusters [76]. Recent discovery of two small molecule inhibitors of CoV assembly that bind the hinge region of M and either stabilize the short form [78] or stabilize a transition intermediate between the short and long forms [79] demonstrates that the dynamic toggling of M between these two forms appears to be critical for virion assembly. Furthermore, M directly interacts with sphingolipids including ceramide-1-phosphate (C1P) in a conformationally selective manner [80], with molecular dynamics simulations revealing stabilization of the short form of M. Ceramide kinase is a cis-Golgi-resident enzyme that converts ceramide into C1P, providing an attractive mechanism for accumulation of M in this compartment.

In a recent study of global subcellular protein reorganization during HCoV-OC43 infection, all seven protein subunits of the COPI complex were mislocalized, which was confirmed by demonstrating that COPE and COPB2 redistribute from a perinuclear location to a broadly dispersed cytoplasmic pattern during infection [81]. Since COPI is responsible for intra-Golgi trafficking, and we have shown that M causes accumulation of protein cargo and cholesterol at the cis-Golgi, we speculate that M may increase the presence of liquid-ordered domains at the cis-Golgi. The formation of COPI vesicles occurs at liquid-disordered regions of the cis-Golgi, and cholesterol is largely excluded from these vesicles [82]. We speculate that recruitment of cholesterol by M could support virus assembly at the expense of COPI recruitment to this site, potentially impairing Golgi protein trafficking and triggering TGN dispersal. This speculation is supported by reports that M binds the Arf1 GTPase that recruits COPI to the cis-Golgi for vesicle formation [83].

The notion of M supporting the formation of rigid ordered lipid domains at sites of CoV assembly is reminiscent of the reports that Spike palmitoylation aids the formation of ordered lipid nanodomains enriched in cholesterol and sphingolipids at ERGIC membranes that aid proper assembly of infectious virions [72]. Importantly, preventing Spike palmitoylation by substituting key cytoplasmic cysteine residues prevented Spike from accumulating at these DRMs, whereas M persisted, suggesting an affinity and independent mechanism for M recruitment to these membranes.

Taken together, our findings provide important new information about how a viral structural protein can moonlight as a host shutoff protein by interfering with subcellular protein trafficking, possibly due to its role in nucleating sites of virion assembly in the early secretory pathway. Considering the newfound focus on M as a target for antiviral small molecules, our study provides a pathway for better understanding the dynamic interactions of M and other viral structural proteins, along with key host sterols and lipids, at these sites of assembly.

## Materials and methods

### Cells and viruses

Human embryonic kidney (HEK) 293T, human adenocarcinoma alveolar basal epithelial A549-ACE2 and human hepatoma Huh7.5 cells were grown in Dulbecco's modified Eagle's medium (DMEM; Thermo Fisher, 11965118) supplemented with heat-inactivated 10% fetal bovine serum (FBS, Thermo Fisher, A31607-01), 100 U/mL penicillin, 100 µg/mL streptomycin, and 2 mM L-glutamine (Pen/Strep/Gln; Thermo Fisher, 15140122 and 25030081). All cells were maintained at 37°C in a 5% $CO_2$ atmosphere. To generate 293T-GFP-ERGIC53 cells, 293T cells were stably transduced with a lentivirus vector encoding GFP-ERGIC53 (pLVX-GFP-ERGIC53-Puro, Addgene #134859), then selected and maintained in 10 µg/mL puromycin (Gibco, A1113803). A549-ACE2 cells were generated by lentivirus transduction of ACE2 as in [84]. Cells were maintained for 48–72 h in 5 µg/mL blasticidin (Thermo Fisher, A1113903) to select for ACE2-expressing cells. Following selection, cells were cultured in DMEM with 10% FBS and Pen/Strep/Gln.

All experiments with severe acute respiratory syndrome coronavirus 2 (SARS-CoV-2) were conducted in the University of Calgary Containment-level 3 (CL3) facility in accordance with the CL3 Oversight Committee and Biosafety Office

regulations. SARS-CoV-2 Toronto-1 variant was propagated in Vero-E6 cells, as in [85]. Briefly, cells were infected at a MOI of 0.01 for 1 h in serum-free DMEM. Following adsorption, cells were maintained in DMEM supplemented with 2% FBS and Pen/Step/Gln at 37°C. Five days post-infection, virus-containing media was centrifuged at 1000 x g for 5 min, aliquoted, and stored at -80°C. SARS-CoV-2 was not passaged beyond passage three. SARS-CoV-2 titers were determined by plaque assay in Vero-E6 cells as in [85] using equal parts 2.4% colloidal cellulose (Sigma, cat # 425244; prepared in sterile H$_2$O) and 2X DMEM (Wisent) supplemented with 1% FBS and Pen/Strep/Gln. 72 h post infection, cells were fixed, stained, and plaques were enumerated.

Stocks of human coronavirus 229E (HCoV-229E; ATCC, VR-740) were propagated in Huh7.5 cells. Cells were infected at a MOI of 0.05 for 1 h in serum-free DMEM. After 1 h, the infected cells were maintained in DMEM supplemented with 2.5% FBS and Pen/Strep/Gln for five days at 33°C. Upon harvest, the culture supernatant was centrifuged at 1000 x *g* for 5 min at 4°C, aliquoted, and stored at -80°C. Stocks of recombinant HCoV-OC43-mClover [86] were propagated in BHK-21 cells. Cells were infected at a MOI of 0.05 for 1 h at 37°C in serum-free DMEM. After 1 h, the infected cells were maintained in DMEM supplemented with 1% FBS and Pen/Strep/Gln until cytopathic effect was complete. Upon harvest, the culture supernatant was centrifuged at 1000 x *g* for 5 min at 4°C, aliquoted, and stored at -80°C. Viral titers were measured using median tissue culture infectious dose (TCID50) assays using the Spearman-Kärber method. Following serial dilution of samples, the appropriate cell line was infected for 1 h at 37°C prior to the replacement of the inoculum for indicated overlay medium and incubated at 37°C: Huh7.5 cells were infected with HCoV-229E in DMEM/2.5%FBS/Pen/Strep/Gln medium, while BHK-21 cells were infected with recombinant HCoV-OC43-mClover in DMEM/1%FBS/Pen/Strep/Gln medium.

To generate lentiviruses for stable transductions, 293T cells were transfected in a 10 cm dish with 3 µg pLVX-GFP-ERGIC-Puro (Addgene #134859) + 2 µg psPAX2 (a gift from Didier Trono; Addgene #12260) + 1 µg pMD2.G (a gift from Didier Trono; Addgene #12259) and 18 µL PEI diluted in Opti-MEM (Gibco, 31985070). After 4 h, the medium was changed to 293T growth medium. After 48 h, supernatants were harvested, cleared by filtration using a 0.45 µm filter, aliquoted, and stored at -80°C until use.

## Chemicals

Tunicamycin (Tm, T7765), Thapsigargin (Tg, T9033), Brefeldin A (BFA, B7651), Cerivastatin (ST, SML0005), and Biotin (B4501) were purchased from Sigma-Aldrich. Tm, Tg, BFA and biotin were solubilized in dimethyl sulfoxide (DMSO) and ST was solubilized in water. All drugs were stored at -80°C. Stock concentrations were diluted to the indicated concentrations in cell culture media.

## Plasmids

All plasmids were purified using the QIAprep Spin Miniprep or QIAfilter Plasmid Midi kits (QIAGEN) and all restriction enzymes were purchased from New England Biolabs (NEB). All SARS-CoV-2 2xStrep tagged constructs used in the ORF library screen (Fig 1) were from the pLVX-SARS-CoV-2 ORF library [42] (kind gifts from Nevan Krogan). The following plasmids were purchased from Addgene: pLDLR-luc (aka: pES7) (#14940, a kind gift from Axel Nohturfft), pTRIP-CMV-tagRFP-FLAG-cGAS and pMSCV-hygro-STING (#86676, #102598, kind gifts from Nicolas Manel), mApple-TNFa-RUSH (#166902, a kind gift from Jennifer Lippincott-Schwartz), pLVX-EF1a-EGFP-ERGIC53-IRES-Puromycin (#134859, a kind gift from David Andrews), and lentiCRISPR v2 (#52961, a kind gift from Feng Zhang). The pCMV-*Gaussia* Luc vector was purchased from ThermoFisher (16147). The pLJM1-Luc2 vector was generated by cloning Luc2 from pGL4.26 (Promega) into pLJM1-B*-Puro [87,88]. The pcDNA3.1+-SARS-CoV-2-Spike (D614) plasmid contains a codon-optimized ORF for Spike from GenBank NC_045512 that was synthesized by GenScript (a kind gift from David Kelvin) then cloned between the *Kpn*I and *Bam*HI sites of pcDNA3.1(+) [87]. To generate pLVX M no tag, M was PCR amplified from the pLVX-M-2xStrep construct from the Krogan library and cloned back into pLVX with *Eco*RI and *Bam*HI. To

generate pLVX E no tag, E was PCR amplified from the pLVX-M-2xStrep construct from the Krogan library and cloned back into pLVX with *Eco*RI and *Bam*HI. All the additional HCoV M proteins (SARS1, MERS, HCoV- OC43, HCoV-HKU1, HCoV-229E, and HCoV-NL63) were synthesized by GenScript in pcDNA3.1 (-) using the GenBank sequences NC_004718.3, NC_019843.3, AY391777.1, NC_006577.2, NC002645.1 and NC_005831.2, respectively. All HCoV M sequences were then cloned into pLVX using *Eco*RI and *Bam*HI. To generate pLJM1-B*-Puro-HA-ATF6 [88] and pLJM1-B*-Puro-HA-ATF6-N, HA-ATF6 and HA-ATF6-N were PCR amplified from pCGN-ATF6 and pCGN-ATF6 (1–373) (Addgene, #11974 and #27173, kind gifts from Ron Prywes) and cloned into pLJM1-B*-Puro with *Nhe*I and *Age*I. To generate the ERSE-luciferase construct pairs of phosphorylated and annealed oligos for 2 ERSE sequences separated by a spacer, were cloned into pGL4.26 (Promega) using *Bgl*II and *Acc*65I. To generate the CMV-*Renilla* construct the *Renilla* luciferase gene was cloned from pcDNA3-RLUC-POLIRES-FLUC (Addgene, #45642, a kind gift from Nahum Sonenberg) into pcDNA3.1(+) using *Nhe*I and *Kpn*I. To generate the pLVX-EV construct pairs of phosphorylated and annealed oligos were cloned into pLVX M no tag, cutting out the M gene with *Eco*RI and *Bam*HI. To generate the pLVX-EGFP, pLVX-mCardinal, pLVX-MEGFP, and pLVX-M-mCardinal constructs either EGFP or mCardinal cloned into pLVX M no tag via cut and paste cloning. The pGL4.26-ISRE (ISRE-luciferase) construct was generated by cloning the 5xISRE response elements from pISRE-luc (Stratagene) with *Bam*HI and *Eco*RI (blunted with Klenow) into pGL4.26 (Promega) with *Bgl*II and *Hin*DIII (blunted with Klenow). The pGL4.26-NFkB (NFkB-luciferase) construct was generated by cloning the 4x NFkB response elements from pNFKB-luc (Stratagene) into pGL4.26 (Promega) using *Nhe*I and *Bgl*II. Primer sequences for PCR amplification and annealed oligos can be found in Table 1.

## Transfections

HEK293T cells were transfected using polyethylenimine MAX (PEI MAX); Linear, MW 40000 (Polyscience, 24765) dissolved in water (pH 7.4). Both plasmids and PEI MAX were diluted in Opti-MEM I (Thermo Fisher, 31985070) and incubated for 5 min, then combined and incubated for 15 min before adding to cell monolayers in antibiotic free 10%FBS/Gln DMEM. For drug treatments, drugs were added to media containing the transfection reaction at indicated times without a media change, excluding the Gaussia-luciferase assays in which media was removed at 24 h post-transfection and replaced with fresh media and BFA for indicated samples.

## CRISPR/Cas9-mediated deletion of human ATF6

HEK293T cells were seeded in a 6-well plate. The following day cells were stably transduced with lentiviruses containing pLentiCRISPRv2-ATF6 plasmids. At 24 h post-transduction cells were selected and maintained in 10 µg/mL puromycin

**Table 1. Primers used for cloning.**

| Target | Sequence (5′ -3′) |
|---|---|
| SARS-CoV-2 M | F-AATTGAATTCACCATGGCCGACTCAAATGG |
| | R-AATTGGATCCTTACTGGACGAGTAAAGC |
| SARS-CoV-2 E | F-AATTCAATTGACCATGTACAGCTTCG |
| | R-AATTGGATCCTTAAACGAGGAGATCAGGCAC |
| ERSE | F-CCCAATCGGCGGCCTCCACGGAGCAGGGCCTTCACCAATCGG CGGCCTCCACGA |
| | R-AGCTTCGTGGAGGCCGCCGATTGGTGAAGGCCCTGCTCCGTGG AGGCCGCCGATTGGGGTAC |
| pLVX EV | F-AATTCCTCGAGACTAGTTCTAGAGC |
| | R-GGCCGCTCTAGAACTAGTCTCGAGG |

(Gibco, A1113803). After expansion of cells under selection, monoclonal ATF6 knockout populations were generated by seeding cells into a 96-well plate at 0.5 cells per well. Wells were screened to ensure populations were expanded from single cells. Individual monoclonal populations were expanded in 6-well plates and screened via immunoblotting for successful deletion of ATF6. A single clone was selected and used throughout the study.

To generate lentiviruses for stable transductions, HEK293T cells were transfected in a 10 cm dish with 3 µg pLentiCRISPRv2-ATF6 + 2 µg psPAX2 (a gift from Didier Trono; Addgene #12260) + 2 µg pMD2.G (a gift from Didier Trono; Addgene #12259) and 18 µL PEI MAX diluted in Opti-MEM I (Gibco, 31985070). After 4 h, the medium was changed to HEK293T growth medium. After 48 h, supernatants were harvested, cleared by filtration using a 0.45 µm filter, aliquoted, and stored at -80°C until use.

To generate pLentiCRISPRv2-ATF6 annealed oligos targeting sequences within the ATF6 gene were cloned into pLentiCRISPRv2 using the following sequences: F-5′-CACCGTTTGCCAATGGCATAAGCGT-3′ and R-5′-ACGCTTATGCCATTGGCAAACGGTG-3′.

### Luciferase reporter assays

HEK293T cells were seeded in 96-well plates coated with poly-L-lysine (Sigma, P2658). The following day cells were transfected with plasmids as indicated in figures. At 24 h post-transfection cells were lysed in 1x Reporter Lysis Buffer (Promega, E397A). For Gaussia-luciferase experiments media was changed at 24 h post-transfection and at 30 h post-transfection supernatant was collected for analysis and cells were lysed as previously indicated. Lysates and supernatant were stored at -80°C until analysis. Once thawed, 10 µL of lysate or supernatant was added to white 96-well plates (Costar, 3917). Measurements of firefly and *Renilla* luciferase were conducted using the Promega Dual Luciferase Kit (Promega, E1910 or E1960), substrates and buffers were prepared as indicated by the manufacturer. Plates were read on a CLARIOstar^PLUS (BMG-Labtech, Serial #430–2826, software version 5.70 R3) and raw data readings were collected in Data Analysis Mars (software version 4.00 R2). pLVX empty vector (EV) or pcDNA3.1(+) were used as controls as indicated for normalization of proteins expressed in the respective vectors.

### Semi-quantitative *XBP1* splicing assay

HEK293T cells were seeded in 12-well plates coated with poly-L-lysine (Sigma-Aldrich, P2658). The following day cells were transfected with pcDNA3.1, pcDNA3.1-SARS-CoV-2-Spike-D614, and pLVX-SARS-CoV-2-M, as indicated. Total RNA from cells was extracted using the RNeasy Plus Mini Kit (QIAGEN, 74134) following the manufacturer's protocol. Synthesis of cDNA was performed using the Maxima H Minus First Strand cDNA Synthesis Kit (ThermoFisher, K1652) using random hexamer primers. A 473 bp PCR product spanning exon/intron boundaries was generated using the *XBP1* forward primer 5′-AAACAGAGTAGCAGCTCAGACTGC-3′ and the *XBP1* reverse primer 5′-TCCTTCTGGGTAGACC TCTGGGAG-3′. The PCR product was digested overnight with PstI-HF to cleave the unspliced *XBP1* product into XBP1u1 and XBP1u2. The digested PCR product was resolved on a 2.5% agarose gel made with 1X Tris-acetate-EDTA and stained with ethidium bromide (Sigma-Aldrich, E1510). The gel was imaged using a ChemiDoc MP Imaging system (Bio-Rad). Quantification of restriction digested PCR products were determined via densitometry using Image Lab 6.1 software (Bio-Rad). The ratio of XBP1s to XBP1u1 and XBP1u2 in all conditions was calculated by dividing XBP1s pixel intensity by XBP1u1+XBP1u2 pixel intensity; these values are then normalized to the EV condition and plotted.

### Immunoblotting

Cell monolayers were washed once with phosphate buffered saline (PBS) and lysed in 2x Laemmli buffer (4% [wt/vol] sodium dodecyl sulfate (SDS), 20% [vol/vol] glycerol, 120 mM Tris-HCl [pH 6.8]). DNA was sheared by repeated pipetting with a 21-gauge needle before adding 100 mM dithiothreitol (DTT), bromophenol blue, and boiling at 95°C for 5 min. Samples were stored at -20°C until analysis. Total protein concentration was determined by DC protein assay

(Bio-Rad, 5000116) against a bovine serum albumin (BSA) standard curve and measured in a 96-well plate format at 750 nm using an Eon (BioTek) microplate spectrophotometer. Equal quantities of 10 μg total protein were loaded in each SDS-PAGE gel, with Color Prestained Protein Standard, Broad Range (NEB, P7719S), and separated at 100 V. Proteins were transferred to polyvinylidene fluoride (PVDF) membranes using the Trans-Blot Turbo RTA Midi 0.2 μm PVDF Transfer Kit (Bio-Rad, 1704273) and a Trans-Blot Turbo Transfer System (Bio-Rad). Membranes were blocked with 5% BSA or 5% skim milk (PERK, BiP, and ATF6 blots) in tris-buffered saline/0.1% [vol/vol] tween-20 (TBS-T) before probing overnight at 4°C with antibodies in 5% BSA in TBS-T raised to the following targets: rabbit anti-SARS-CoV-2 S1 RBD (Elabscience, E-AB-V1006, 1:2000), rabbit anti-SARS-CoV-2 M (Novus Biologicals, NBP3–05698, 1:2000), rabbit anti-SARS-CoV-2 E (abbexa, abx226552, 1:2000), rabbit anti-PERK (Cell Signaling Technologies (CST), #5683, 1:2000), mouse anti-CHOP (CST, #2895, 1:1000), mouse anti-XBP1s (CST, #12782, 1:1000), rabbit anti-BiP (CST, #3177, 1:1000), mouse anti-HA (CST, #2367, 1:1000), mouse anti-ATF6 (AbCam, ab122897, 1:1000), rabbit polyclonal anti-Caveolin (BD Biosciences, C13630, 1:1000), rabbit anti-ERGIC-53 (Sigma, E1031, 1:2000), and rabbit anti-β-actin (CST, #8457, 1:1000). Membranes were washed with TBS-T and incubated with HRP-linked secondary antibodies for 1 h at room temperature in 5% BSA in TBS-T. Secondary antibodies used: anti-rabbit, HRP-linked (CST, #7074, 1:3000), and anti-mouse, HRP-linked (CST, #7076, 1:3000). Blot were developed with Clarity ECL chemiluminescence reagent (Bio-Rad, 170–5061) or Clarity Max ECL chemiluminescence reagent (Bio-Rad, 170–5062) for PERK, ATF6 and BiP blots. All blots were imaged using a ChemiDoc MP Imaging System (Bio-Rad). Molecular weights were determined using the Color Prestained Protein Standard, Broad Range (NEB, P7719S). Molecular weights in kDa are indicated on the left of blot images. Images were analyzed using Image Lab 6.1 (Bio-Rad). Images were cropped and annotated using Affinity Designer (Serif).

**Immunofluorescence microscopy**

HEK293T cells were seeded on #1.5 coverslips (Paul Marienfeld GmbH & Co. KG, 0117580) coated with poly-L-lysine (Sigma-Aldrich, P2658). The following day cells were transfected or infected with indicated plasmids or viruses, respectively. To harvest, the cells were washed once with PBS and fixed with 4% paraformaldehyde (PFA; Electron Microscopy Services, 15710) in PBS for 15 min at room temperature. Coverslips were blocked and permeabilized in staining buffer (1% human serum (Sigma-Aldrich, 4552) heat-inactivated at 56°C for 1 h, 0.1% Triton X-100 in PBS) for 1 h at room temperature. Coverslips were stained overnight at 4°C with the following antibodies as indicated: mouse anti-SARS-CoV-2 M (R&D Systems, MAB10696), rabbit anti-GM130 (CST, 12480S, 1:3000), rabbit anti-TGOLN2 (BETHYL, A304-434A, 1:200) and mouse anti-dsRNA J2 (SCICONS, RNT-SCI-10010200, 1:500). The following day the coverslips were washed three times with PBS, then stained with secondary antibodies: goat anti-mouse-555 (Invitrogen, A21422, 1:1000), anti-mouse-647 (Invitrogen, A21463, 1:1000) goat anti-rabbit-647 (Invitrogen, A21244, 1:1000), anti-rabbit-488 (Invitrogen, A21441, 1:1000) donkey anti-rabbit-555 (Invitrogen, A31572, 1:1000), and chicken anti-mouse-488 (Invitrogen, A21200, 1:1000) in staining buffer for 1 h at room temperature in the dark. Coverslips were washed with PBS three times, and then counterstained for 5 min with Hoescht 33342 (ThermoFisher, 62249). Coverslips were then mounted on cover glass (Fisher Scientific, 12–550)) using ProLong Gold anti-fade reagent (ThermoFisher, P36930). Z-stacks were imaged on a Zeiss LSM880 and processed into maximum intensity projections using Zen Black (Zeiss).

To image intracellular cholesterol localisation, we used the D4H-mCherry cholesterol probe, which is a plasmid encoding the domain 4 (D4) of theta-toxin produced by *Clostridium perfringens* with a D434S substitution (D4H) that lowers the threshold for binding cholesterol [73,89–91]. When imaging cells transfected with this probe, cells were fixed with 4% PFA and then blocked with 1% human serum as above, but we omitted Triton X-100 to minimize membrane disruption, however membranes were sufficiently disrupted by fixation to allow for immunostaining of GM130 using an increased concentration of antibody (rabbit anti-GM130, 1:1000). For BODIPY-480/508-Cholesterol (BCh2) (Cayman Chemicals, 24618–500) staining, the dye was prepared as a 1 mg/mL solution in DMSO. Cells were transfected with mCardinal or

M-mCardinal constructs as described above except at 4h post-transfection medium was replaced with 10% FBS DMEM containing 2 µg/mL of dye, which was filtered at 0.2 µm. Cells were fixed 24h after transfection with 4% PFA, mounted using Prolong Gold, and imaged on a Zeiss LSM880 as described above.

### Retention using selective hooks (RUSH) cargo sorting assay

HEK293T cells were transfected with mApple-TNFa-RUSH, along with pLVX EV, or pLVX M as indicated. Cells were treated with biotin (Sigma, B4501) at time of transfection (24 h treatment) or at 20 h post-transfection. At 24 h post-transfection cells were fixed, stained and imaged as described above.

### Isolation of detergent resistant membranes (DRM) from cells

DRMs were isolated essentially as described in [92]. Briefly, HEK293T cells were seeded into 3-wells of a 6-well cluster dish and transfected with pLVX-M or EV the following day as described above. Cells were then washed once with ice-cold PBS, then once-with TNE buffer (150 mM NaCl, 2 mM EDTA, 50 mM Tris–HCl, pH 7.4) then scraped and washed once more in TNE buffer. The cell pellet was then resuspended in 200 µL TNE with complete EDTA-free Protease Inhibitor Cocktail (Roche) then homogenized with 25-G needle (25-strokes). 180 µL of the lysed cell suspension was transferred to a new tube with 20 µL of freshly prepared 10% Triton X-100, mixed by inversion, and incubated on ice for 30 min. Then 400 µL of Optiprep (60% iodixanol; Sigma) added, mixed, and the resulting 600 µL was moved to an ultra-centrifuge tube for a S-55S (ThermoFisher) ultracentrifuge rotor and carefully overlaid with 1.2mL 30% iodixanol-TNE then 300 µL of TNE without iodixanol. The tubes were then transferred to the S-55S rotor and centrifuged at 55,000 RPM for 2 h. Two 1 mL fractions were then removed from the tube and used for immunoblot analysis as described above.

### Data management and analysis

Graphing and statistical calculations were performed using GraphPad Prism for macOS v10.3.1. Figures were prepared using Affinity Designer v1.10.8 (Serif). Raw microscopy images were processed and intensities standardized using ImageJ2 v2.16.0/1.54n. Merged microscopy images were uploaded to https://amsterdamstudygroup.shinyapps.io/ezre-verse/ for image inversion. Additional images for biological replicates for immunoblotting and immunofluorescence micros-copy experiments are available in the Dryad Database [93].

## Supporting information

**S1 Fig. Knockout of ATF6 from HEK293T cells confirming selectivity of ERSE-luciferase assay. (A)** HEK293T cells were transduced with lentiviruses encoding pLentiCRIPSRv2-ATF6 constructs to knockout the ATF6 gene. At 24 h post-transduction cells were selected in 10 mg/mL of puromycin. Cells were seeded in a 96-well plate to generate mono-clonal populations and expanded under puromycin selection. Selected monoclonal populations were seeded into 6-well plates, lysates were harvested 48 hours post-seeding and stored at -20C prior to immunoblotting. * indicates the knockout clonal population selected for further experimental use. **(B)** Wild-type HEK293T or ATF6-KO cells were transfected with Spike or treated with Tg for 1h prior to harvest. (n = 3 ± SD, statistical significance was determined by one-way ANOVA with Fisher's LSD test. ****, adj. P < 0.00001 relative to EV WT or EV KO.).
(TIFF)

**S2 Fig. Human coronavirus infections lead to reduced structure and dispersal of both the *cis*-Golgi and the *trans*-Golgi network.** Confocal immunofluorescence images of GM130 (*cis*-Golgi), or TGOLN2 (*trans*-Golgi) in coronavirus infected cells. **(A)** HEK293T cells were infected with HCoV-OC43-mClover at an MOI of 0.05 then fixed at 24 h post-infection. Infected cells were infected with HCoV-229E at an MOI of 0.05 then fixed at 24 h post-infection. Infected cells

were identified with the J2 monoclonal antibody that binds dsRNA. Maximum intensity projections are presented. 100X magnification, scale bar = 10 μm. Representative images of three independent experiments.
(TIFF)

**S1 Data. Source data for all graphs.** This table presents source data for Figs 1A-G, 2D, 2F, 3A-D, 5C, 6C, 6E, and S1B.
(XLSX)

## Acknowledgments

We thank the following colleagues for their assistance this work: Dr. Roy Duncan and Nichole McMullen for the use of the CLARIOstar^PLUS plate reader. We thank Dalhousie University Core Facility managers Dr. Gerard Gaspard (Cellular Molecular Digital Imaging) and Dr. Christopher Hughes (Biological Mass Spectrometry) for expert technical support. We thank Dr. Nevan Krogan (UCSF) for the generous gift of a collection of lentiviral vectors expressing SARS-CoV-2 ORFs [42]. We thank Dr. Greg Fairn (Dalhousie) for the generous gift of the pmCherry-D4H plasmid. The following plasmids were gifts provided through Addgene: pLVX-EF1a-EGFP-ERGIC53-IRES-Puromycin (David Andrews, https://www.addgene.org/134859/), mApple-TNFa-RUSH (Jennifer Lippincott-Schwartz, https://www.addgene.org/166902/); pLDLR-Luc (aka: pES7) (Axel Nohturfft, https://www.addgene.org/14940/); pTRIP-CMV-tagRFP-FLAG-cGAS and pMSCV-hygro-STING (Nicola Manel, https://www.addgene.org/86676/, https://www.addgene.org/102598/); lenti-CRISPRv2 (Feng Zhang https://www.addgene.org/52961/); psPAX2 (Didier Trono, https://www.addgene.org/12260/); pMD2.G (Didier Trono, https://www.addgene.org/12259/); pCGN-ATF6 (Ron Prywes, https://www.addgene.org/11974/); pCGN-ATF6 (1-373) (Ron Prywes, https://www.addgene.org/27173/).

## Author contributions

**Conceptualization:** Taylor M Caddell, Eric S Pringle, Craig McCormick.

**Formal analysis:** Taylor M Caddell, Eric S Pringle, Craig McCormick.

**Funding acquisition:** Jennifer A Corcoran, Eric S Pringle, Craig McCormick.

**Investigation:** Taylor M Caddell, Rory P Mulloy.

**Project administration:** Eric S Pringle, Craig McCormick.

**Supervision:** Jennifer A Corcoran, Eric S Pringle, Craig McCormick.

**Validation:** Taylor M Caddell, Rory P Mulloy, Eric S Pringle.

**Visualization:** Taylor M Caddell, Rory P Mulloy, Eric S Pringle.

**Writing – original draft:** Taylor M Caddell, Eric S Pringle, Craig McCormick.

**Writing – review & editing:** Taylor M Caddell, Jennifer A Corcoran, Eric S Pringle, Craig McCormick.

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
