## [Decision Letter · Decision Letter 0]

22 Jan 2026

PPATHOGENS-D-25-02753

Coronavirus M proteins disperse the trans-Golgi network and inhibit anterograde protein trafficking in the secretory pathway

PLOS Pathogens

Dear Dr. McCormick,

Thank you for submitting your manuscript to PLOS Pathogens. After careful consideration, we feel that it has merit but does not fully meet PLOS Pathogens's publication criteria as it currently stands. Therefore, we invite you to submit a revised version of the manuscript that addresses the points raised during the review process.

The reviewers appreciated the important new insights into the mechanism of modulation of the cellular membrane trafficking pathways upon coronavirus infection, but also indicated  that significant improvements are required in the result presentation and interpretation

We look forward to receiving your revised manuscript.

Kind regards,

George A. Belov, PhD

Academic Editor

PLOS Pathogens

Alexander Gorbalenya

Section Editor

PLOS Pathogens

Sumita Bhaduri-McIntosh

Editor-in-Chief

PLOS Pathogens

orcid.org/0000-0003-2946-9497

Michael Malim

Editor-in-Chief

PLOS Pathogens

orcid.org/0000-0002-7699-2064

Additional Editor Comments:

Consider using singular "Coronavirus M protein" in the title and elsewhere. It applies to all coronaviruses and avoids "M proteins" that could be confusing.

Please adhere to the acronym usage as agreed in coronavirus field: NSP -> nsp, hCoV -> HCOV

Journal Requirements:

At this stage, the following Authors/Authors require contributions: Taylor Caddell, Rory Mulloy, Jennifer Corcoran, Eric S Pringle, and Craig McCormick. Please ensure that the full contributions of each author are acknowledged in the "Add/Edit/Remove Authors" section of our submission form.

https://journals.plos.org/plospathogens/s/submission-guidelines#loc-parts-of-a-submission

5) We notice that your supplementary Figures are included in the manuscript file. Please remove them and upload them with the file type 'Supporting Information'. Please ensure that each Supporting Information file has a legend listed in the manuscript after the references list.

6) When completing the data availability statement of the submission form, you indicated that you will make your data available on acceptance. We strongly recommend all authors decide on a data sharing plan before acceptance, as the process can be lengthy and hold up publication timelines. Please note that, though access restrictions are acceptable now, your entire data will need to be made freely accessible if your manuscript is accepted for publication. This policy applies to all data except where public deposition would breach compliance with the protocol approved by your research ethics board. If you are unable to adhere to our open data policy, please kindly revise your statement to explain your reasoning and we will seek the editor's input on an exemption. Please be assured that, once you have provided your new statement, the assessment of your exemption will not hold up the peer review process.

Reviewers' Comments:

Reviewer's Responses to Questions

Part I - Summary

Reviewer #1: This merit of this well -written submission is with its demonstration that coronavirus membrane (M) proteins impede host protein anterograde transport in the exocytic pathway. The findings bring greater insights into published findings showing that coronavirus infections corrupt host exocytic organelles. The demonstration that M proteins operate globally in halting protein cargo transit at the cis Golgi is a significant contribution. The additional novelty – as communicated by authors – is that M proteins can now be viewed as more than the fundamental components of the virion. They can now be viewed as participants in host cell remodeling processes that both limit antiviral responses and foster development of proviral virion budding sites.

This submission begins by testing UPR activation by SARS2 viral proteins. The results showed that three arms of the UPR pathway were activated by spikes (S) [and by chemically-induced protein misfolding] but only the ATF6 arm was reduced by co-expressed M. ATF6 activation requires ATF6 transit to Golgi – hence the authors developed the hypothesis that M blocks ER to Golgi transport. [Note that this hypothesis diverges from the UPR-centric beginning of the paper – however the story is clearly developed by the authors.] The hypothesis was then tested in several ways; M reduced SREBP and STING activation (both operate from Golgi), M reduced Gluc secretion, M accumulated in ERGIC along with cholesterol, RUSH assays showed a reporter protein transport from ER to ERGIC, and no further, when M proteins were present, and M dispersed the later TGN organelles. The results generally supported the conclusion that M accumulation in ERGIC halts protein transport at the ERGIC. This appears to be a worthy report that will be appreciated by cell biologists and virologists.

Reviewer #2: Caddell et al. present data supporting a model in which the Membrane (M) protein of SARS-CoV-2 (and other coronaviruses) disrupts the trans-Golgi network (TGN) such that secretory membrane trafficking is impaired. Consequently, the unfolded protein response (UPR) mediated by ATF6 is inhibited. The authors also show that this arm of the UPR as well as those mediated by PERK and IRE1 are triggered by Spike, but only the activation of ATF6 is inhibited my M, consistent with their model in that only ATF6 requires secretory transport for activation. The inhibition of ER-localized sensors by M extends to SREBP2 and STING, again consistent with the proposed mechanism, although sensing by these cellular proteins of coronavirus proteins or replication was not shown. Lastly, the authors note modulation of cholesterol at the cis-Golgi by M and propose that this is the likely mechanism by which M disrupts the TGN. Overall, the study is well-presented, and the conclusions follow the data. As noted below, some of the imaging data is not fully convincing.

Reviewer #3: The study examines effects of Cornona virus proteins on ER stress,  golgi structure and secretory function.  Specifically, they look for effects on UPR activation, golgi dispersement and protein trafficking in the secretory pathway. They found that Spike protein triggered UPR whereas M protein inhibited the ATF6 pathway and blocked antergrade trafficking of UPR sensor proteins.  They observe cholesterol accumulation in the cis Golgi and dispersing of the trans Golgi suggesting the virus could be a way to interfere with host response to infection while maintaining ability to use cell to make nascent virions.    The study expands on other observations of corona virus proteins altering ER stress response and identifies M protein as a regulator of the ATF6 pathway or more generally altering the anterograde trafficking of proteins in the secretory pathway.  One of the concerns is the use of an enzymatic activity (luciferase) of a reporter protein to measure changes in proteostasis where the activity could be due to protein misfolding and not transcriptional changes.  Assessment of general cell viability would also be useful to rule out that decreases are not due to cytotoxicity.  Abbreviations should be clearly defined especially the use of “EV” which was interpreted to be extracellular vesicle prep from virally infected cells.  Overall, the findings appear novel and expand our understanding of corona virus M protein in altering cellular physiology.

Part II – Major Issues: Key Experiments Required for Acceptance

Reviewer #1: 1. The vast majority of experiments involved plasmid overexpression in HEK293 cells. While the results were convincing, they may not reflect events taking place in natural infection contexts. For example, suppression of UPR, SREBP2, STING, could come from nsp1 or other viral proteins, moreso than M. Perhaps the discussion section should include communication of the caveats associated with the results coming from reductionist assays in HEK293 cells.

2. Key figures 4-6 show convincing results but fragmentation of trans Golgi is hard to see. Also, increased cholesterol densities in GM130+ punctae are difficult to notice. One question : Why are the IFA images in color only in the merged panels, and not in the panels depicting single signals? It would help to see the single signals in color. Alos, scale bars should be added to the panels.

3. While most of the submission is very clear, the discussion paragraph on lines 331-344 confuses this reviewer. This paragraph infers a relationship between broadly dispersed COPI, M accumulation in cis-Golgi, liquid ordered membranes in the cis-Golgi, TGN dispersal, and spike palmitoylation. There are many components here and this reviewer struggles with the alleged connections. This paragraph is potentially valuable and therefore it deserves additional clarifying text.

4. One further comment related to #3; as authors speculate on mechanisms by which M proteins accumulate in cis-Golgi (linkage to ceramide-1-P was brought up) and impair cargo trafficking through Golgi, has this recent paper describing M-Arf1 interactions been considered (https://doi.org/10.1038/s41467-025-61431-8 ) ?

Reviewer #2: 1) Figure 4, which shows microscopically the localization of M with respect to cellular markers and the dispersal of the TGN by M, has no quantitation. At the least, co-localization data regarding M and the markers of the cis- and trans-Golgi should be generated. I wish I had a good suggestion for how to quantify TGN-dispersal, but perhaps the authors can think of a way to do it. One way might be to pick similar juxatanuclear areas in cells and determine the signal intensity (i.e., the signal density) of the TGN marker, since it appears to drop when M is expressed.

2) Figure 5, which uses the RUSH assay, is not convincing. Peripheral puncta appear present in the no-biotin condition, rendering the interpretation difficult. The choice of TNF as an indicator protein might not have been optimal. Wouldn't a protein like VSV-G, which would reach the plasma membrane when released from the ER, have been a better indicator?

3) Figure 6, which examines the effect of M on cholesterol in the cis-Golgi, is also not fully convincing or clear. In panel C, what are the four individual graphs representing? The control of rotating one channel is interesting, but why not obtain a Pearson's coefficient of correlation between the signals? Panel D lacks quantitation.

4) The authors do not provide evidence that perturbations of cholesterol cause by M are causative with regard to TGN dispersal; both can be true but not causally related.

Reviewer #3: What is EV?  Extracellular vesicles?  Not defined or referenced.  It says “transfected with EV” but then refers to “EV” cells elsewhere.   Also not clear what EV is from the methods/materials. Given this reviewer was delayed in reviewing this paper for variety of reasons, I will provide additional comments without fully understanding EV and assuming it is related to extracellular vesicles from infected cells.

In Figure 1, was there any independent verification that alterations of ATF6 pathway occurred using an endogenous marker (protein or RNA)?  Not necessarily needed for all proteins but for M effects?

Figure 1, could the decrease in luciferase be due to luciferase misfolding and not a reduction in the transcription of the ERSE reporter since protein folding and processing may be altered? For example, consider examining Luc mRNA expression by qPCR to show that it is also reduced for at least the strongest effect.

The Tg responses in Figure 1F seem relatively low.  Also, could the low activity once again be due to M causing misfolding of the reporter?  Or high toxicity levels to the cells?  What is the Tg/Tm response alone?  And was the EV and M response alone ran in parallel?  These data are difficult to interpret without the controls. In Fig 1F, why are doses presented high to low (opposite to typical dose response and counterintuitive).

Figure 2D, as the authors state in methods this is a semiquantitative assay and it is not clear what is being calculated in the right panel.  Also not clear is how many replicates were used (states n=3, is that 3 wells or 3 independent experiments?

Figure 2F, the results (line 159) states the cells were ATF6 KO cells but figure legend states HEK293T cells.  Which one was used?  What is the reference for ATF6 KO cells?

Figure 3.  Same concerns about protein folding of the different luciferases since activity is the endpoint measure and protein folding is being disrupted.

Quantitation of colocalization should be considered.  Pearson’s or Manders’ correlation plots should be straight forward with data presented.

For Figure 6, it is not clear whether the location of sensor or the location of cholesterol is being affected by M protein.  Is there at difference from M-eGFP and MeGFP (to clarify there isn’t a M-EGFP fusion).  Both are mentioned in the results section.

Part III – Minor Issues: Editorial and Data Presentation Modifications

Reviewer #1: (No Response)

Reviewer #2: 1) Figure 1E: by eye one would think that ORF10 is activating ATF6 and that M is inhibiting that, but those values are labelled as "ns" (not significant). Is that correct?

2) In supplemental Figure 2 it does not appear that infection with hCOV-229E is dispersing the TGN.

3) Lines 262-3, "...we did not find that M could similarly alter the lipid composition of the ERGIC." What is the basis for this statement in the data presented?

4) Line 900 has a grammatical error: "remained" could be replaced by "remainder" to correct this.

Reviewer #3: Check document for spelling, grammar, font, etc.

PLOS authors have the option to publish the peer review history of their article (what does this mean?). If published, this will include your full peer review and any attached files.

Do you want your identity to be public for this peer review? For information about this choice, including consent withdrawal, please see our Privacy Policy.

Reviewer #1: No

Reviewer #2: No

Reviewer #3: No

Figure resubmission:
---

## [Decision Letter · Decision Letter 1]

25 Mar 2026

Dear Dr. McCormick,

We are pleased to inform you that your manuscript 'Coronavirus M protein disperses the trans-Golgi network and inhibits anterograde protein trafficking in the secretory pathway' has been provisionally accepted for publication in PLOS Pathogens.

Best regards,

George A. Belov, PhD

Academic Editor

PLOS Pathogens

Alexander Gorbalenya

Section Editor

PLOS Pathogens

Sumita Bhaduri-McIntosh

Editor-in-Chief

PLOS Pathogens

orcid.org/0000-0003-2946-9497

Michael Malim

Editor-in-Chief

PLOS Pathogens

orcid.org/0000-0002-7699-2064

Reviewer Comments (if any, and for reference):

Reviewer's Responses to Questions

Part I - Summary

Reviewer #1: The authors have improved their manuscript by responding appropriately to reviewer concerns.

Reviewer #2: In this revision and in response to the reviewers' comments, the authors have improved the primary data presented as well as its description and discussion. In summary, the manuscript presents a cogent and interesting story of how the M protein of coronaviruses can inhibit the innate sensing of infection by disrupting the Golgi.

Reviewer #3: The authors present data that support the COV membrane (M) selectively inhibits the ATF6 arm of the UPR and M accumulates with cholesterol at the cis-Golgi where it inhibits anterograde trafficking and dispersing the trans-Golgi network (TGN). Suggests that the virus could interfere with host response requiring trafficking of new proteins but not interfere with egress of COV virions.

Part II – Major Issues: Key Experiments Required for Acceptance

Reviewer #1: (No Response)

Reviewer #2: None.

Reviewer #3: Authors have adequately addressed concerns.

Part III – Minor Issues: Editorial and Data Presentation Modifications

Reviewer #1: (No Response)

Reviewer #2: None.

Reviewer #3: (No Response)

PLOS authors have the option to publish the peer review history of their article (what does this mean?). If published, this will include your full peer review and any attached files.

Do you want your identity to be public for this peer review? For information about this choice, including consent withdrawal, please see our Privacy Policy.

Reviewer #1: No

Reviewer #2: No

Reviewer #3: No

---

## [Editor Report · Acceptance letter]

Dear Dr. McCormick,

We are delighted to inform you that your manuscript, "Coronavirus M protein disperses the trans-Golgi network and inhibits anterograde protein trafficking in the secretory pathway," has been formally accepted for publication in PLOS Pathogens.

Best regards,

Sumita Bhaduri-McIntosh

Editor-in-Chief

PLOS Pathogens

orcid.org/0000-0003-2946-9497

Michael Malim

Editor-in-Chief

PLOS Pathogens

orcid.org/0000-0002-7699-2064